# Monkeypox Outbreak in Peru

**DOI:** 10.3390/medicina59061096

**Published:** 2023-06-06

**Authors:** Max Carlos Ramírez-Soto

**Affiliations:** 1Centro de Investigación en Salud Pública, Facultad de Medicina Humana, Universidad de San Martín de Porres, Lima 15011, Peru; maxcrs22@gmail.com or c20330@utp.edu.pe; 2Facultad de Ciencias de la Salud, Universidad Tecnológica del Peru, Lima 15046, Peru

**Keywords:** monkeypox outbreak, public health, emergency, Peru

## Abstract

Monkeypox (Mpox) is a zoonotic disease caused by the *Orthopoxvirus* monkeypox virus (MPXV). Since 1970, outbreaks of MPXV have occurred in several Sub-Saharan African countries. However, from May 2022 to April 2023, recent outbreaks of Mpox occurred in several countries outside of Africa, and these cases quickly spread to over 100 non-endemic countries on all continents. Most of these cases were found in the region of the Americas and the Europe region. In Latin America, the highest all-age Mpox rates per million inhabitants were in Peru, Colombia, Chile, and Brazil. Given its global impact, Mpox was declared as an international Public Health Emergency by WHO in July 2022. MPXV infection disproportionately affects men who have sex with men and members of the HIV-infected population. Vaccination is the current strategy for controlling and preventing Mpox in high-risk groups. In this context, Peru has the fourth-highest number of Mpox cases in Latin America and faces significant challenges in disease control. Because of this, in this review, we discuss the epidemiology, public health indicators, and prevention of Mpox in the 2022 Peru outbreak so that health authorities can join forces to control MPXV transmission.

## 1. Introduction

Mpox virus (MPXV) is a zoonotic infection belonging to the genus *Orthopoxviruses*, family *Poxviridae*. It is transmitted via droplet exposure and/or direct contact with contagious materials [1,2]. Since 1970, outbreaks of Mpox have occurred in several Sub-Saharan African countries, primarily in the Democratic Republic of Congo (DRC), Cameroon, the Central African Republic (CAR), Liberia, the Republic of Congo (ROC), and Nigeria. Since then, MPXV has been restricted to these countries [3,4]. However, recent outbreaks of MPXV have occurred in countries outside of Africa, and a high number of cases have quickly spread to almost every continent [2,5,6]. Most cases occur in young men, with a significant proportion of them being men who have sex with men (MSM) and other high-risk groups, whose transmission mainly occurs by close human-to-human contact and sexual contact. [7,8]. In this review, we discuss the epidemiology, transmission and population at risk, public health indicators, and prevention of Mpox in the 2022 Peru outbreak.

## 2. COVID-19 Pandemic in Peru

Despite the early implementation of a national lockdown and other restraint measures to prevent SARS-CoV-2 transmission, the COVID-19 pandemic severely impacted the Peruvian population, resulting in a high death rate and COVID-19 incidence, excess COVID-19 deaths, and an excess of death from all causes [9,10,11]. Possible explanations for the poor outcomes of the COVID-19 pandemic are a fragmented healthcare system, a lack of specialized human resources to tackle the pandemic, gaps in infrastructure, a lack of molecular tests (first wave), a lack of intensive care unit beds, a lack of essential drugs, the use of medications without evidence of their efficacy, a lack of leadership from health authorities, and a pandemic response that was directed toward hospitals and not primary healthcare [12,13]. These factors contributed to exacerbating the problem. The Peruvian health system was severely affected, and its health facilities were overfilled with COVID-19 patients. In addition, an urgent need for improvement became evident since non-COVID-19 patients could not access regular healthcare services, which had an indirect impact on the population’s health [13]. Peru was severely affected by the COVID-19 pandemic, with more than 4,500,000 cases and 220,000 deaths reported as of 21 April 2023 [14]. However, after the implementation of a vaccination program to combat COVID-19, the burden of disease and death has decreased considerably.

## 3. Epidemiology

The first case of human Mpox was reported in 1970 in the DRC. Since then, it has spread to West and Central Africa, and the number of cases has been on the rise [15,16,17]. From 2000 to 2015, there were outbreaks reported in the DRC and Nigeria [3]. The case fatality rate (CFR) in these outbreaks was 8.7%. For the Central African clade and West African clade, the CFR was 10.6 and 3.6%, respectively [3]. Since 2003, the spread outside of Africa has been related to import and travel to endemic countries, which occasionally resulted in outbreaks [18,19,20]. According to the WHO, Mpox in 2022 was considered endemic in several African countries [21]. In the current global outbreak, the first case of Mpox was reported in May 2022 in the United Kingdom [22]. Since then, MPXV has spread to 111 countries around the world. Given its global impact, Mpox was declared as an international Public Health Emergency by WHO in July 2022 [6].

In terms of cumulative number, from 1 January 2022 to 25 April 2023, there were a total of 87,113 reported cases of Mpox in 111 countries worldwide, including a total of 130 deaths [22]. Most of these cases were found in the region of the Americas (59,220 cases) and the Europe region (25,881 cases) (Figure 1). In Latin America, Peru is the country with the fourth most reported cases, after Mexico, Colombia, and Brazil. In Europe, Spain and France are the countries with the most reported cases (Figure 2) [22].

To date (27 April 2023), seventeen countries had an all-age cumulative rate of Mpox cases > 50 cases per million inhabitants, including Gibraltar, Spain, Peru, Portugal, the United States of America, Luxembourg, Colombia, Chile, Netherlands, Belgium, Malta, Puerto Rico, Switzerland, France, the United Kingdom, Brazil, and Panama (Figure 3).

In Latin America, the highest all-age Mpox rates were in Peru (113.9 cases per million inhabitants), Colombia (79.8 cases per million inhabitants), Chile (74.9 cases per million inhabitants), and Brazil (50.9 cases per million inhabitants) (Figure 3).

## 4. Transmission and Population at Risk

While the animal reservoirs of the MPXV are little known, transmission can occur from animal to human (zoonotic transmission) and from human to human [2,4,21]. During the 2022 Mpox outbreak, it was reported that person-to-person transmission occurred by several routes including respiratory secretions, such as droplets generated when an infected person coughs or sneezes; contact with infected fluids or skin lesions of an MPXV-infected person; percutaneous transmission (such as cuts, abrasions, or puncture wounds); or indirect contact through fomites [7,8,22]. The risk factors for acquiring the MPXV include high-risk sexual behavior (multiple sexual partners or anonymous sexual partners) in the Americas and European countries and living in forested areas in African countries [7,8,18,21,23]. This outbreak affected primarily gays, bisexuals, and MSM [7,8,22]. Although women were the least-affected population in the 2022 outbreak, a global case series found that between cis women, including non-binary individuals and trans women, 61% and 89% of them acquired Mpox through sexual contact [24].

## 5. Sexually Transmitted Infections Concomitant

Reports indicate that people with Mpox may also have concomitant sexually transmitted infections (STIs). In the current global outbreak of human Mpox, high rates of HIV infection [22] and other STIs were reported among individuals with Mpox [7,8,22]. In previous Mpox outbreaks in Nigeria, there were also concurrent HIV infections [25,26]. In eight USA jurisdictions, the HIV prevalence in Mpox-infected people was 38%, and 41% were diagnosed with one or more other STIs. These STIs included *N. gonorrhoeae*, *C. trachomatis*, and syphilis in the Mpox patients with and without a diagnosis of HIV infection [27]. In Spain, herpes simplex virus I/II, *N. gonorrhoeae*, *C. trachomatis*, and syphilis were detected in Mpox patients [28]. In Italy, STIs more frequent were *N. gonorrhoeae* and *M. genitalium* [29]. In Germany, *N. gonorrhoeae*, *C. trachomatis*, syphilis, and *Mycoplasma* were reported [30]. In Mexico, HIV, syphilis, and some cases of chronic hepatitis C were also identified in patients with Mpox [31]. In Latin America, HIV, syphilis, genital herpes, chlamydia, and gonorrhea were reported in Brazil [32], while in Peru, only HIV and syphilis were reported in patients with Mpox infection [33,34].

## 6. MPXV Outbreak in Peru

Soon after the first descriptions of Mpox outbreaks in several countries around the world, on 26 May 2022, the Centro Nacional de Epidemiología, Prevencion y Control de Enfermedades (CDC Peru), Peruvian Ministry of Health issued a health alert to health establishments about the risk of imported Mpox cases in the national territory [35]. Subsequently, the government began implementing its control plan for the Mpox outbreak on 7 June, which included health establishments preparing and responding to possible cases [36]. Despite the implementation of a control plan for the Mpox outbreak, we knew that the introduction of MPXV was imminent because of the large numbers of national and international travelers, combined with problems in the Peruvian health system.

Peru’s first Mpox case was diagnosed on June 15 in the department of Lima. By mid-July, community transmission was occurring, and the country had neither a sufficient response nor the contact-tracing capacity to contain MPXV. Because of this, on 30 September 2022, the Peruvian Ministry of Health published a new version of the technical health standard for the prevention and management of patients affected by MPXV. This rule aims to protect the population at high risk or affected by MPXV (probable or confirmed cases), including guidelines for community preventive measures and care in health facilities. Despite the implementation of a control plan against Mpox, the disease was almost certain to spread rapidly in Peru because of the same factors that influenced poor outcomes during the COVID-19 pandemic.

Since the first case of Mpox infection was found in Peru (15 June 2022), community transmission was occurring. Beginning on week 36, 2022, the cases have decreased by week 12, 2023 (Figure 4A). Up to 25 April, 3800 human Mpox cases and ten deaths were reported by the CDC Peru [34]. As in other countries [7,8,27,32], most of these cases occurred in young men (96.1%) and MSM (56%). The most-affected population is people living with HIV (55%) and those who receive antiretroviral therapy (85%) (Figure 4B) [34]. According to the WHO (to 19 October 2022), among Mpox cases, 25,718 cases were reported in MSM [22]. This shows that the current spread of MPXV has disproportionately affected MSM, suggesting the amplification of transmission through sexual networks. Although the Mpox outbreak has predominately affected gay men, Mpox is not a “gay disease” [37]. Likewise, the WHO reports that among Mpox cases, approximately 50% of cases are HIV-positive people [22]. As in Peru, several reports in Europe, the United States, and Brazil describe a high rate of HIV infection among Mpox cases [7,27,28,29,30,31,32,33,38,39]. Given the transmission route, MPXV could still find other transmission networks, and it could also start to spread among sex workers, their clients, and other population groups [37].

Most cases and the highest rates were recorded in large cities, such as Lima, Callao, La Libertad, and Arequipa (Figure 5A,B) [34]. Figure 5C shows the number of infections by province, which allows us to understand the spread of infections between cities, and how the Mpox cases spread from the province of Lima, where the first cases were reported, to the various provinces over time. With this information can better understand the progression and dynamics of the Mpox outbreak and identify high-risk areas.

In the late 1970s, smallpox was eradicated through vaccination, and in Peru, as in other countries, the resurgence of MPXV is likely due to smallpox vaccines being phased out over the last four decades, as well as population growth and densely overpopulated areas, which facilitate virus spread [6]. Because of this, numerous individuals are now susceptible to MPXV. The current outbreak of Mpox in Peru also might be driven by changes in human behavior, such as the relaxation of the COVID-19 pandemic prevention measures, including the resumption of international and national travel. Sexual interactions are also associated with social events and large gatherings. In this context, we are currently bearing witness to a monkeypox outbreak in Peru, with the fourth-highest number of cases of Mpox infection in Latin America [22].

## 7. Mpox Outbreak and Global Health Security Index

The Global Health Security (GHS) indicators may also help explain the current outbreak in Peru. The GHS Index is a tool developed by the Johns Hopkins Center for Health Security and the Nuclear Threat Initiative, in collaboration with the Economist Intelligence Unit. It assesses 195 countries’ capabilities in detecting, preventing, and responding to epidemics and pandemics [40]. The GHS Index evaluates various factors such as detection, prevention, reporting, health system, rapid response, international norms compliance, and risk environment. The GHS Index aims to identify gaps in preparedness and encourage countries to strengthen their health security capacities [40]. According to the 2021 GHS Index report, all countries, including Peru, are unprepared for future threats such as epidemics and pandemics; 38.9 is the average overall score in the 2021 GHS Index [40]. The United States is the country with the highest score on the 2021 GHS Index (75.9), and Somalia is the country with the lowest score (16.0) [40]. Peru has a 54.9 Index score (32/195 rank), with low indicators for prevention, detection, response, health, norms, and risks (Figure 6) [40]. Therefore, Peru remains vulnerable to future outbreaks due to failures in quarantine and isolation policies and disease control. During the COVID-19 pandemic, some studies showed that the overall GHS Index was correlated with SARS-CoV-2 infections and COVID-19 deaths [41,42]. In the same vein, countries in the Americas region with different GHS Index scores have different rates of Mpox cases. For example, Peru, the United States, Colombia, Chile, and Brazil are the countries with the highest overall GHS indicators, and they have the highest case rates of Mpox cases in the region (Figure 7). Even the improved GHS indicators regarding the capability to detect, prevent, and respond to epidemics and pandemics cannot guarantee success in controlling the Mpox outbreak; however, these indicators can help contextualize the Mpox outbreak in Latin America and Peru.

## 8. MPXV Vaccination

We currently have vaccines that offer protection against MPXV, but their availability is limited in some countries [43]. Despite this, mass vaccination is not recommended for Mpox. The WHO provides many interim recommendations on vaccination and immunization against Mpox to prevent the spread of human-to-human MPXV infection, mainly in groups at high risk of exposure [44]. Primary preventive vaccination (pre-exposure) is recommended for the groups that have a high-risk of exposure to MPXV, including bisexuals, gays, MSM, sex workers, laboratory professionals who work with viruses of the genus *Orthopoxviruses*, etc. [44]. Additionally, there is a recommendation for the post-exposure preventive vaccination for contacts of Mpox cases. It is advised to administer the Mpox vaccine within four days of the first exposure. In the absence of symptoms, post-exposure vaccination can be given up to 2 weeks after the initial exposure [44]. In the United States and the United Kingdom, the JYNNEOS vaccine (also known as IMVAMUNE), produced by Bavarian Nordic, is used in high-risk groups to prevent Mpox [43,45], but this vaccine was not available in Peru. This vaccine is administered subcutaneously to individuals 18 years of age or older (two doses with intervals of 4 weeks). It can be used pre-exposure to prevent infection in high-risk exposure groups, or also post-exposure (ideally up to 4 days after exposure) [46,47].

In the current Mpox outbreak, vaccination programs against Mpox are focused on high-risk exposure groups. Globally, the Mpox vaccine is being administered in European countries, the United States, and Latin America, while African countries have limited access to vaccines. [21]. The Peruvian government announced its first batch of an MPXV vaccine in October 2022. These were 5600 of a total of 9800 doses acquired by the country. In the second delivery, 4200 doses arrived in November 2022 [48]. The Mpox vaccination process initiated two phases. The first phase is in people with HIV, and the second phase is in vulnerable populations such as MSM, bisexuals, transgender women, and workers and sex workers [49]. In that sense, the government should work to ensure timely access to vaccines for high-risk groups, including vulnerable populations such as MSM, bisexual men, and HIV-positive people in areas with a high number of Mpox cases and a high amount of risk contact.

## 9. Reflections and Recommendations

There are several lessons to be learned from outbreaks and previous pandemics in response to the global Mpox outbreak. First, the investment in healthcare capacity and science is essential to building robust and timely responses [6,50]. Second, the implementation of screening policies, contact tracing, and increased awareness of Mpox in the general population can help in the early detection of cases and disease control. Third, the preparedness for new outbreaks and pandemics should be focused on planning and long-term investments in public health. This includes strengthening surveillance systems, establishing early warning systems, and enhancing laboratory capabilities for the rapid and accurate diagnosis of new pathogens. Additionally, capacity building is important, including training healthcare professionals, scientists, and emergency responders. Finally, the findings of the CDC Peru show that some risk groups are disproportionately affected. Thus, to contain the outbreak, special attention should be placed on MSM, people with HIV, and other groups at elevated risk levels, and Mpox vaccination programs should target these high-risk groups. Even if more vaccines do arrive in Peru, behavioral changes will be needed in groups with a high-risk of exposure to MPXV.

## Figures and Tables

**Figure 1 medicina-59-01096-f001:**
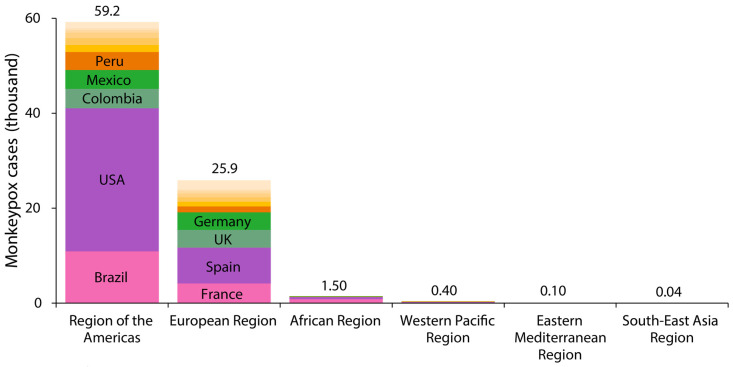
Stacked bar chart of regional distribution of Mpox cases for 2022–2023 (Source: Mpox, WHO) [22]. This figure was built using the cases of Mpox data from the WHO.

**Figure 2 medicina-59-01096-f002:**
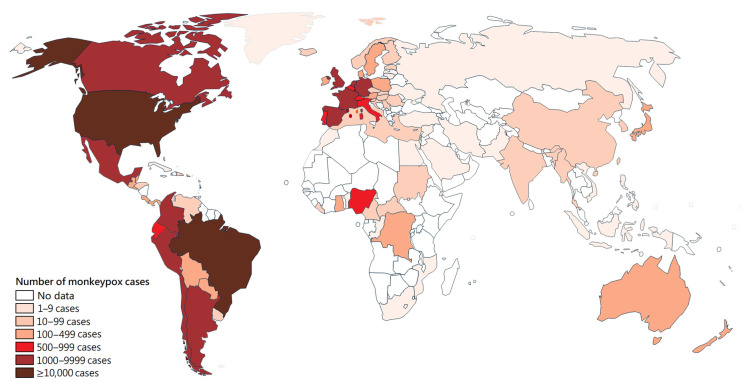
Global distribution of Mpox cases for the period 2022–2023. This figure was built using the cases of Mpox data from the WHO (Source: Monkeypox outbreak—WHO; accessed on 25 April 2023) [22].

**Figure 3 medicina-59-01096-f003:**
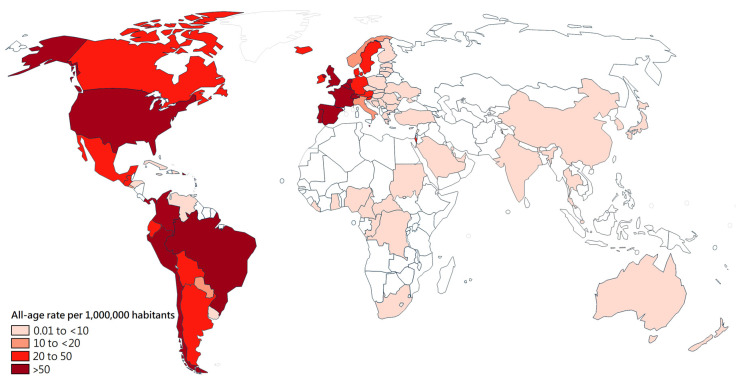
Global distribution of estimated Mpox rate for the cumulative period 2022–2023. Calculated rate by the number of cases reported for the cumulative period 2022–2023 (≥5 cases) in a certain country [22] divided by the total population of that country of that same year as reported by https://data.worldbank.org/indicator/SP.POP.TOTL (accessed on 25 April 2023).

**Figure 4 medicina-59-01096-f004:**
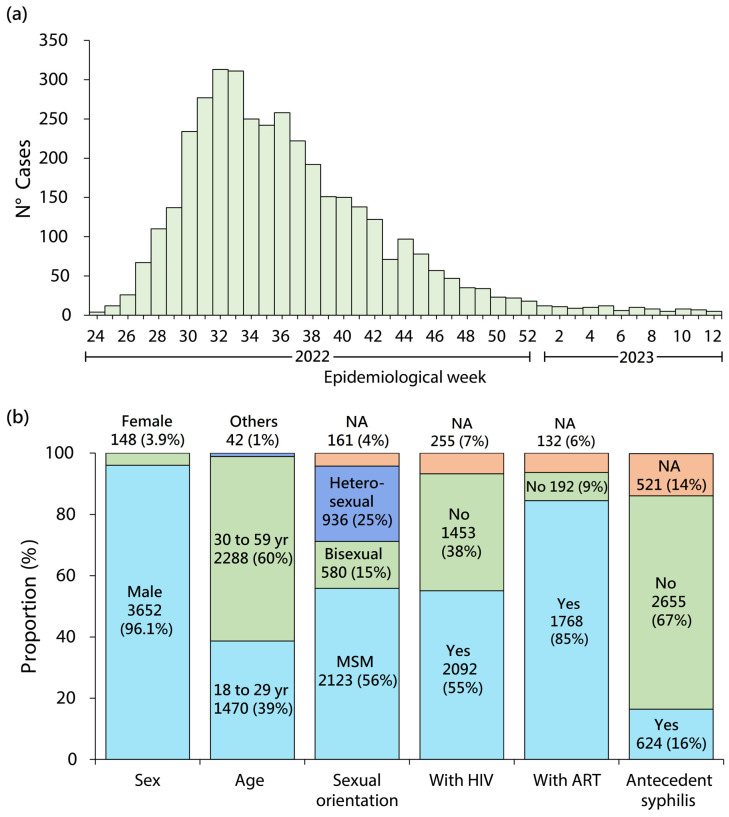
Human monkeypox cases registered in Peru from 15 June to 25 April 2022 (Source: Centro Nacional de Epidemiología, Prevencion y Control de Enfermedades (CDC Peru), Peruvian Ministry of Health) [34]. (**a**) Weekly distribution of human Mpox cases. (**b**) Features of human monkeypox cases. NA: not available; MSM: men who have sex with men; ART: antiretroviral therapy. These figures were built using the cases of Mpox data from the CDC Peru, Peruvian Ministry of Health.

**Figure 5 medicina-59-01096-f005:**
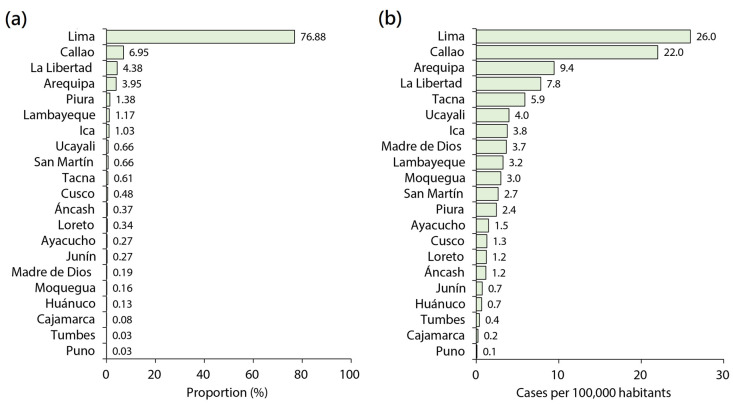
Distribution of Mpox cases in Peru from 15 June 2022 to 23 April 2023 (Source: Centro Nacional de Epidemiología, Prevencion y Control de Enfermedades (CDC Peru), Peruvian Ministry of Health) [34]. (**a**) Proportion of Mpox cases by department. (**b**) Mpox rate by department. (**c**) Number of Mpox cases by province. These figures were built using the cases of Mpox data from the CDC Peru, Peruvian Ministry of Health. Mpox rate was estimated between the number of Mpox cases for the cumulative period 2022–2023 in a certain region [34] divided by the total population of that region per 100,000 inhabitants.

**Figure 6 medicina-59-01096-f006:**
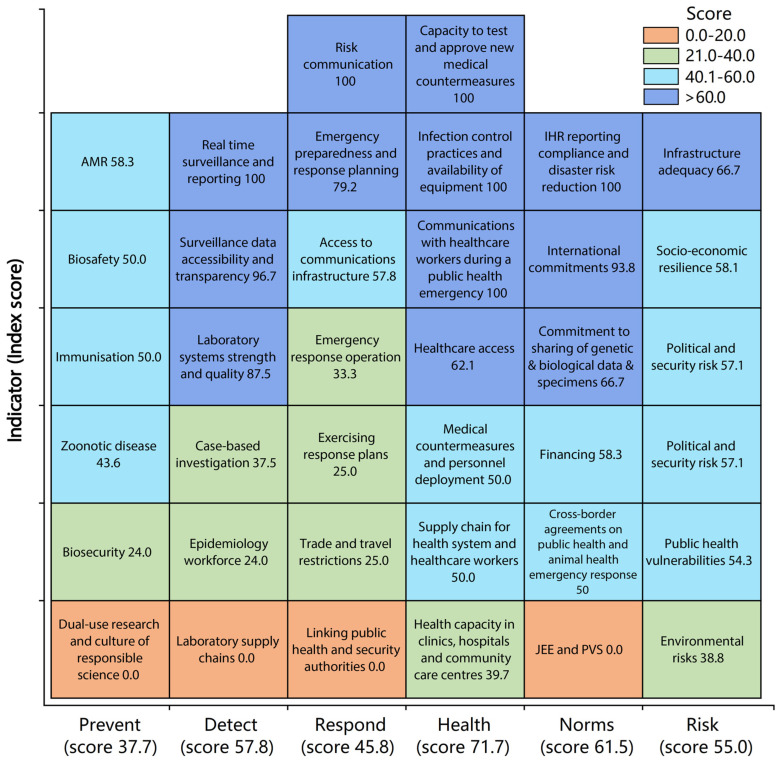
Global Health Security Index in Peru, 2021 [40]. This figure was built using the Global Health Security Index data.

**Figure 7 medicina-59-01096-f007:**
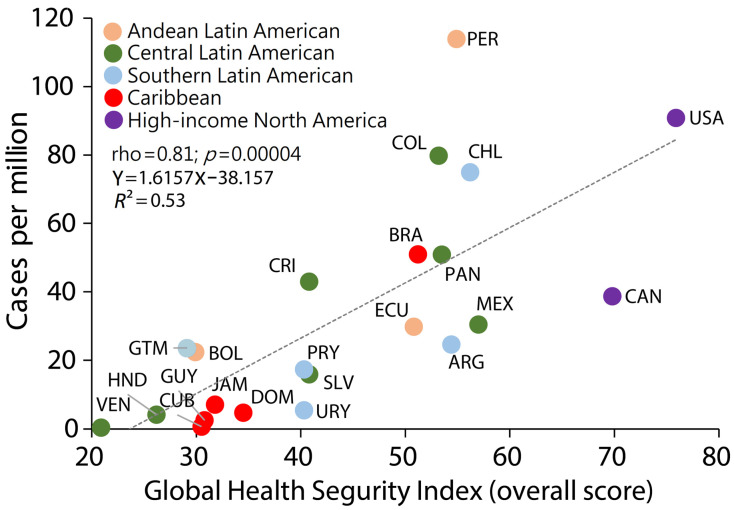
Relationship between Global Health Security (GHS) Index score [40] and Mpox cases in Americas region (data updated to 25 April 2023) [22]. This figure was built using the Global Health Security Index data. Spearman’s analysis shows countries with higher GHS Index had higher Mpox case rates per million inhabitants.

## Data Availability

Not applicable.

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
