# Peer review of "Monkeypox Outbreak in Peru"

_medicina, 2023, doi:10.3390/medicina59061096_

Round 1
Reviewer 1 Report
The review work presented by Max Carlos Ramírez-Soto titled “Monkeypox Outbreak in Peru: Reflections after the COVID-19 Pandemic” is well-written, clear, and easy to read. The topic is interesting; therefore, it adds clustered information to the monkeypox concern in Peru that is the fourth-highest number of MPXV cases in Latin America.
Author Response
Comment: The review work presented by Max Carlos Ramírez-Soto titled “Monkeypox Outbreak in Peru: Reflections after the COVID-19 Pandemic” is well-written, clear, and easy to read. The topic is interesting; therefore, it adds clustered information to the monkeypox concern in Peru that is the fourth-highest number of MPXV cases in Latin America.
Response: Thank you for your comments.
Reviewer 2 Report
The author conducted a comprehensive review about monkeypox in Peru. The whole manuscript is well-written, so I just have one minor suggestion.
1. The discussion of monkeypox is enough and there is no direct relationship betwen monkeypox and COVID-19. May remove the discussion about COVID-19 from title and the final part.
Author Response
We thank the Reviewer for your comments and constructive criticism, we believe that the quality of our manuscript has been significantly improved. We have revised our paper in a point-by-point manner.
Comment: The author conducted a comprehensive review about monkeypox in Peru. The whole manuscript is well-written, so I just have one minor suggestion. The discussion of monkeypox is enough and there is no direct relationship between monkeypox and COVID-19. May remove the discussion about COVID-19 from title and the final part.
Response: Thank you for your comments. The manuscript has been corrected.
Reviewer 3 Report
This review compares public health measures for COVID-19 and Mpox in Peru and shows the importance of approaching high-risk populations in common. It is commendable in showing the need to approach MSM, HIV-infected persons, and other high-risk groups during a Mpox epidemic and to prioritize Mpox vaccination for these populations. On the other hand, the explanation on COVID-19 is practically limited to L236-249, which is a discrepancy between the title and the article's content. Therefore, the phrase "after COVID-19" should be deleted from the title. Figure 4A shows Peru's total number of infections, which does not allow us to understand the spread of infections between cities. If possible, the number of cases should be broken down by province to see how the disease spread from the province of Lima, where the first cases were reported, to the various provinces. The population of a province also has a significant impact, so the population ratio should also be taken into account and shown.
Author Response
This review compares public health measures for COVID-19 and Mpox in Peru and shows the importance of approaching high-risk populations in common. It is commendable in showing the need to approach MSM, HIV-infected persons, and other high-risk groups during a Mpox epidemic and to prioritize Mpox vaccination for these populations.
We thank the Reviewer for your comments and constructive criticism, we believe that the quality of our manuscript has been significantly improved. We have revised our paper in a point-by-point manner.
Comment 1. On the other hand, the explanation on COVID-19 is practically limited to L236-249, which is a discrepancy between the title and the article's content. Therefore, the phrase "after COVID-19" should be deleted from the title.
Response 1: Thank you for your comments. The manuscript has been corrected.
Comment 2. Figure 4A shows Peru's total number of infections, which does not allow us to understand the spread of infections between cities. If possible, the number of cases should be broken down by province to see how the disease spread from the province of Lima, where the first cases were reported, to the various provinces. The population of a province also has a significant impact, so the population ratio should also be taken into account and shown.
Response 2. Thank you for your comments. A map of the distribution of cases at the province level has been included.
Round 2
Reviewer 3 Report
The authors appropriately reflect the points made and the argument is extremely clear.